# Variation in the Local Grey Mullet Populations (*Mugil cephalus*) on the Western Pacific Fringe

**DOI:** 10.3390/genes15101280

**Published:** 2024-09-29

**Authors:** Chien-Hsien Kuo, Sin-Che Lee, Shin-Yi Du, Chao-Shen Huang, Hung-Du Lin

**Affiliations:** 1Department of Aquatic Bioscience, National Chiayi University, Chiayi 60004, Taiwan; hagkuo@mail.ncyu.edu.tw; 2Institute of Cellular and Organismic Biology, Academia Sinica, Taipei 115, Taiwan; sclee.academiasinica@gmail.com (S.-C.L.); dudu@gate.sinica.edu.tw (S.-Y.D.); 3Fisheries Research Institute Kaohsiung Branch, Kaohsiung 806, Taiwan; cshuang.academiasinica@gmail.com; 4The Affiliated School of National Tainan First Senior High School, Tainan 701, Taiwan

**Keywords:** allozyme, gene flow, geographic variation, *Mugil cephalus*, GPI

## Abstract

**Background**: Understanding population genetic structures is crucial for planning and implementing conservation programmes to preserve species’ adaptive and evolutionary potential and thus ensure their long-term persistence. The grey mullet (*Mugil cephalus*) is a globally distributed coastal fish. Its populations in waters surrounding Taiwan on the western Pacific fringe are divided into at least two stocks (migratory and residential), but questions remain regarding their genetic divergence and gene flow. **Methods and Results**: To cast more light on this, allozyme variations at 21 presumptive gene loci of 1217 adult grey mullets from 15 localities in Japan, Taiwan and mainland China, and four gene loci from 1470 juveniles from three localities in Taiwan were used to investigate patterns of genetic variation. The mean expected heterozygosity (He) was 0.128—ranging from 0.031 (Matsu) to 0.442 (Kaoping)—and the mean observed heterozygosity (Ho) was 0.086—ranging from 0.017 (Kaohsiung) to 0.215 (Kaoping). Both AMOVA and the high overall mean FST of 0.252 indicated enormous genetic differentiation among populations and the positive mean value of FIS was 0.328, indicating a deficiency of heterozygotes. PCoA indicated that the samples of *M. cephalus* could be split into three groups and STRUCTURE analysis showed that all individuals were grouped into three genetic clusters. The results of mutation-drift equilibrium tests did not suggest that the populations experienced any recent genetic bottleneck. The results from all localities in the present investigation showed significant change in the *GPI-A* genotype frequencies with latitudes—e.g., increases in *GPI-A*135/135* homozygote frequencies and *GPI-A*100/100* frequencies were highly correlated with latitudinal cline. All migratory populations with the *GPI-A* genotype were almost exclusively the *GPI-A*100/100* homozygote. During the life history of *M. cephalus*, the *GPI-A*100/135* heterozygote frequency significantly decreases with age. **Conclusions**: Based on these data, we suggest that each *GPI-A* genotype represents trait combinations of higher fitness in some portions of the environment. Furthermore, the genotypic frequencies change in accordance with life stages, suggesting that selection occurs throughout the life span.

## 1. Introduction

Complex geological processes, climatic history, and diverse coastal habitats across different regions of the world create opportunities that shape the current phylogeographic patterns of marine organisms [1,2]. The northwest Pacific margin waters include the Sea of Japan, Yellow Sea, East China Sea, and South China Sea. Ocean currents, ancestral habitat discontinuity, and climatic constraints in the northwest Pacific may play an important role in shaping the contemporary genetic and population structures of marine organisms [3]. During the Pleistocene glacial periods, the Taiwan Strait emerged due to the decline of sea levels, and it has acted as a barrier to the movement of marine organisms across the two sides of this strait [4,5]. These historical geological events, combined with complex coastal habitats, have been linked to significant genetic barriers between two evolutionary lineages of marine species due to behavioural or oceanographic constraints, such as in Chinese black sleeper (*Bostrychus sinensis*) [6] and shimofuri goby (*Tridentiger bifasciatus*) [7]. Conversely, some species exhibit low genetic structure, likely due to higher migration rates, as seen in cutlassfish (*Lepturacanthus savala*) [8], prawn (*Macrobrachium japonicum*) [9], and hairtail (*Trichiurus japonicus*) [10]. The genetic structure of fish populations has attracted a great deal of attention, not just because of basic interest in the evolution of biotic organisms but mainly because of its importance for fishery management.

The grey mullet (*Mugil cephalus*) is a globally distributed marine fish, inhabiting coastal waters and estuaries in tropical and subtropical seas between latitudes 42° N and 42° S [11,12,13]. Spawning occurs offshore near the surface, with buoyant eggs hatching approximately 48 h after fertilization. Larvae are dispersed across the continental shelf by ocean currents, spending the first 2–3 months in a planktonic stage. During this phase, they grow to a standard length (Ls) of 16–20 mm and form dense schools that migrate toward inshore waters and estuaries [14]. Young recruits first appear in the surf zone before moving into the shallow areas of sounds, bays, and estuaries. Here, juveniles (40–69 mm Ls) spend their first year in waters with salinities ranging from 0 to 35‰ [15]. Adult grey mullets primarily feed on detritus and reach sexual maturity in their third year, at which point they form large migratory schools [16,17].

In Taiwan’s coastal waters and estuaries, two types of grey mullet are observed: migratory and resident [18]. The migratory type originates from the northern East China Sea, travelling along mainland China’s coast to the Taiwan Strait during the spawning season [19]. Migratory adults and juveniles then return to the mainland coast, aided by the Kuroshio Current, while juveniles remain in Taiwanese estuaries until late April. In contrast, the resident type inhabits Taiwanese estuaries year-round, with minimal migration. The grey mullet’s life history involves both passive and active dispersal mechanisms, primarily along the shoreline, which likely shape population subdivisions. Understanding the population dynamics and genetic structure of this species is crucial, given its significance to coastal fisheries and aquaculture in many countries worldwide [13].

Previous genetic work revealed that the populations of *M. cephalus* have two to three different lineages in the western Pacific fringe [18,20]. Analyzing the temporal patterns of lineages can provide insight into the contemporary and historical genetic connectivity of *M. cephalus*. For example, a COI phylogenetic tree revealed three lineages: the NWP1 lineage has a northward distribution from Taiwan to Russia, the NWP2 lineage is distributed along the warm Kuroshio Current, and the NWP3 lineage has a distributional range that suggests tropical affinities [18]. There are two life histories of *M. cephalus* (migratory and residential populations) in Taiwan, and gene flow occurs between them. Population genetic structures differ with estimated genetic distances and the genetic markers used. For example, Sun et al. [20] clustered *M. cephalus* in the China Sea into two groups based on mitochondrial DNA—one for the populations from the Bohai and East China Seas and the other from the Yellow and South China Seas—while Jamandre et al. [21] indicated that *M. cephalus* in the northwest Pacific belongs to two highly divergent lineages, with the inferred population structure being closely associated with the distribution of both lineages. According to Livi et al. [22], *M. cephalus* appears to consist of highly isolated populations characterized by specific mitochondrial lineages. 

Huang et al. [23] previously employed the *GPI-A* locus as a genetic marker to distinguish between at least two grey mullet stocks—migratory and residential—within the western Pacific fringe, indicating that natural selection might influence allelic variation. In this study, we applied allozyme electrophoresis to investigate the potential genetic structure of grey mullet populations around Taiwan. Our objectives were threefold: first, to assess genetic divergence and gene flow among western Pacific populations of *Mugil cephalus*; second, to evaluate whether local populations are subject to selective pressures; and third, to determine the timing of selection during the grey mullet’s life cycle, specifically whether genotypic mortality occurs progressively or results from intense selection at a particular developmental stage. 

## 2. Materials and Methods

In total, 1291 adult grey mullets 15.1–54.0 cm Ls were collected from 15 localities in three regions: (1) Japan’s archipelago: Nagasaki (NA); (2) the coast of mainland China: Shanghai (SH), Tachen (TC), and Matsu (MS); and (3) Taiwan’s main island: Tanshui (TS), Dashi (DS), Hualien (HL), Wuchi (WC), Tadu (TD), Peimen (PM), Anping (AP), Chiding (CD), Kaohsiung (KS), Kaoping (KP), and Tapong (TP). An additional 1470 juveniles were collected from three localities in Fulung (FLJ), Tanshui (TSJ), and Linbien (LBJ) in Taiwan estuaries (Figure 1). The population of *M. cephalus* in waters adjacent to Taiwan migrate annually from the feeding grounds in coastal waters of China to offshore waters of both southwestern and northeastern Taiwan to spawn in December at 3–4 years old [24]. From December to January, samples from the spawning population of *M. cephalus* collected from Wuchi (WC), Anping (AP), Chiding (CD), and Kaohsiung (KS) were defined as being migratory. These specimens were frozen at −75 °C immediately after being transported live to the laboratory. All studies in animals were conducted by ethical committee guidelines and approved by the Animal Research and Ethics Committee of department of Aquatic Bioscience, National Chiayi University (Taiwan) (Protocol Number: D20200318-01), and the study was carried out in compliance with the ARRIVE guidelines. All surgeries were performed under MS-222 anesthesia, and all efforts were made to minimize suffering. These specimens were frozen at −75 °C immediately after being transported live to the laboratory. After examining the enzyme tissue-specific distribution among the brain, eye, heart, skeletal muscle, liver, gill, kidney, and gonad, we chose to use the skeletal muscle and heart in the subsequent experiments. We used the following buffers: TVB [25], TC 7.0, TC 8.0 [20], and LiOH [26]. Tissue was homogenized with 2 to 3 volumes of buffer (0.1 mM Tris-HCl pH 7.0, 1 mM Na_2_ EDTA, and 0.05 mM NADP^+^) and centrifuged at 17,000× *g* for 40 min at 4 °C. Electrophoresis was run on a 12% (*w*/*v*) starch gel (Sigma Chemical Company, St. Louis, MO, USA) and stained with the recipes following the methods of Jean et al. [27]. Thirteen loci nomenclature followed recommendations from Shaklee et al. [28]. Alleles at each locus encoded were identified according to the migration mobility of the protein; the most common allele present was scored 100.

Allozyme analysis is a technique used in population genetics and evolutionary biology, but it has limitations, such as limited detection of genetic variability and susceptibility to environmental influences [29]. Allelic and genotypic frequencies of examined loci in *M. cephalus* populations were obtained by counting phenotypes directly from the gels. The mean number of alleles per population (Na), the proportion of polymorphic loci at the 95% level (P_95_), the observed heterozygosity (Ho), and the expected heterozygosity (H_E_) were computed by ARLEQUIN 3.5 [30]. All loci were tested for proximity to the Hardy–Weinberg equilibrium (HWE), and all pairwise combinations of loci were tested for linkage disequilibrium using ARLEQUIN 3.5 [30]. The allelic richness (A_R_) and inbreeding coefficient (*F*_IS_) for each population were estimated using FSTAT v.2.9.3 software [31] and GenePop Web Version 4.0.10 [32], respectively. F-statistics (*F*_IS_, *F*_IT_, and *F*_ST_) for the polymorphic allozyme loci were estimated according to Weir and Cockerham [33] using the software FSTAT v.2.9.3 [31]. 

To identify candidate loci potentially influenced by the selection of *M. cephalus* based on the allozyme loci, we implemented the Fdist approach of Beaumont and Nichols [34] in LOSITAN v. 1.6 [35] and the BayeScan v. 2.1 [36] software based on the *F*_ST_ outlier approach. In LOSITAN, a neutral distribution of *F*_ST_ with 50,000 iterations was simulated with a false discovery rate of 0.1 and confidence interval of 0.95. All loci putatively identified by the LOSITAN program were removed from the dataset to generate a panel of allozyme loci, conform to assumptions of neutrality, and avoid misleading signals in population structure. We used the BayeScan software [36] and the R function “plot_bayescan” (https://cmpg.unibe.ch/software/BayeScan/download.html; accessed on 21 November 2023) to detect loci under selection. Pairwise *F*_ST_ values and a hierarchical analysis of molecular variance (AMOVA), based on allele frequency information, were calculated to evaluate the amount of population genetic structure using ARLEQUIN Ver 3.5 with 1000 permutations [30]. For the hierarchical AMOVA, the populations were grouped according to seven scenarios: (1) no groups (Scenario I, N = 14); (2) the Taiwan, Japan, and mainland China groups (Scenario II, N = 3); (3) the phylogeny results (Scenario III, N = 3), including NA—WC, CD, KS, AP, SH, TC—and MS—TS, DS, TD, HL, KP, PM and TP; (4) the Taiwan and other population groups (Scenario IV, N = 2); (5) 22 samples when *GPI 100/100* and *GPI 135/135* are treated separately (Scenario V, N = 2); (6) the Taiwan, Japan, and mainland China groups when GPI loci are removed (Scenario VI, N = 2); and (7) Taiwan and Japan in a migratory group (Scenario VII, N = 2). Tests for isolation by distance (IBD) were carried out in order to determine whether genetic differentiation increased with geographic distance using a Mantel test (10,000 permutations) in ARLEQUIN 3.5 [30]. We used the information on the coordinates from Google Earth for measuring the distance between the sampling sites.

To determine the relationships among the populations based on Nei [37], the genetic distance between all pairs of populations was examined. A dendrogram was also created using the unweighted pair grouping method with the arithmetic mean (UPGMA) method with bootstrap values calculated using POPULATIONS ver.1.2. [34], which was viewed on TreeView. Based on the allozyme loci, both principal component analysis (PCoA) and a Bayesian clustering method were used to explore the genetic clusters of *M. cephalus* and its relatives. A principal component analysis (PCoA) based on the standardized covariance of genetic distances between populations was performed using the program GenAlEx v 6.5 [38]. Bayesian assignment tests were applied to estimate the number of genetic clusters and to evaluate the degree of admixture among them using STRUCTURE v2.3.3 [39] based on microsatellite data. An estimation of the number of subpopulations (K) was completed using 10 independent runs with K = 1–25 (assuming no prior population delineation information) at 1,000,000 MCMC repetitions combined with a 100,000-repetition burn-in period. The posterior probability of each K value was calculated using an estimated log-likelihood, and the likelihood ratio was tested to determine the optimal numbers of subgroups. To obtain the most appropriate number of genetic groups in our dataset, the most likely K-value was determined in Structure Harvester Web 0.6.94 [40,41] using the log posterior probability of the data for a given K, Ln Pr (XjK) [42]. To determine whether the *M. cephalus* populations had experienced a recent shrinkage in effective population size using the software BOTTLENECK 1.2.02 [43], the observed distribution of allele frequency was compared to that of a population in a mutation-drift equilibrium assuming the IAM (infinite allele model).

## 3. Results

### 3.1. Genetic Variations and Population Structures of Adult Grey Mullets

A total of 2761 grey mullets—1291 adults and 1470 juveniles—were investigated based on 21 loci scored from 13 isozyme markers (*mAAT*, *CK-A*, *GPI-A*, *GPI-B*, *IDH-A*, *IDH-B*, *LDH-A*, *LDH-B*, *sMDH-A*, *MPI*, *PGDH*, *PGM-A*, and *PGM-B*). Their genotypic frequencies are presented in Appendix A. The number of samples (Na), number of alleles per population (Ab), allelic richness (A_R_), expected heterozygosity (H_E_), observed heterozygosity (H_O_), and inbreeding coefficient (*F*_IS_) for each sample are shown in Table 1. The average number of alleles in each population ranged from 2.000 (DS, HL, CD, AP, WC, TC) to 3.400 (TSJ, juveniles) (average = 2.367). The mean allelic richness for each population ranged from 1.035 (CD) to 1.382 (SH) (average = 1.188). The mean expected heterozygosity was 0.128 (ranging from 0.031 (MS) to 0.442 (KP)), and the mean observed heterozygosity was 0.086 for each sample (ranging from 0.017 (KS) to 0.215 (KP)). The positive *F*_IS_ (heterozygote deficiencies) observed in most populations ranged from 0.000 (CD and TC) to 0.698 (KS), except those from the WC (−0.017) and MS (−0.007) populations (Table 1).

The results of Weir and Cockerham’s (1984) estimates of F-statistics showed that the highest values of *F*_IS_ (0.643) and *F*_IT_ (0.652) occurred at the locus *mAAT*, whereas the highest value of *F*_ST_ (0.253) occurred at the *PGDH* locus (Table 2). The jackknife resampling procedure allowed us to calculate a standard deviation of global F-values over loci (global *F*_IS_ = 0.452 ± 0.124; global *F*_IT_ = 0.578 ± 0.130; global *F*_ST_ = 0.218 ± 0.044). Overall, the jackknifing and the bootstrapping over the locus (for a 95% and a 99% confidence interval, respectively) showed higher *F*_ST_ levels than the *F*_IS_ or *F*_IT_ levels (Table 2). The marked positive global *F*_IS_ indicates a likely heterozygote deficiency within *M. cephalus* samples.

The number of alleles for each of the 13 isozyme loci (average = 3.00) ranged from two (*CK-A*, *IDH-A*, *IDH-B*, *LDH-B*, *MDH-A*, and *PGM-B*) to five (*GPI-A*) in *M. cephalus* (Table 2). The average allelic richness per locus was 1.249, ranging from 1.003 (*MDH-A*) to 2.415 (*GPI-A*). The observed heterozygosity (H_O_) was 0.029, ranging from 0.000 (*MDH-A*) to 0.185 (*GPI-A*), and the expected heterozygosity (H_E_) was 0.043, ranging from 0.000 (*MDH-A*) to 0.296 (*GPI-A*). The mean value of *F*_IS_ was 0.328, and five of the thirteen studied isozyme loci (*mAAT*, *GPI-A*, *MDH-A*, *PGM-B*, *PGDH*) had a positive *F*_IS_ value, higher than that predicted by the HWE.

Appendix A shows the pairwise *F*_ST_ values (−0.010 (between DS and HL) to 0.608 (between KP and MS), with a mean value of 0.252). The pairwise *F*_ST_ values among the NA (Nagasaki) and other samples were significantly different in all pairwise comparisons (Appendix A). The analysis of variance (AMOVA) was applied to test the probable factors shaping genetic structure according to geographical barriers. The AMOVA results indicated that most of the genetic variation was within individuals, i.e., one group (80.70%), three groups (Scenario I, 60.91%), three groups (Scenario II, 59.64%), and four groups (Scenario III, 75.62%) (Table 3). When the populations were divided into one group, three groups (Scenario I), three groups (Scenario II), and four groups (Scenario III), 0%, 27.20%, 34.71%, and 19.85% of the total variation was found among the group divisions, respectively (Table 3). No significant correlation was found between geographic distance and genetic differentiation of populations when all populations were included in the IBD analyses (r = 0.193, *p* = 0.107). Furthermore, the significant correlation between pairwise genetic and geographic distances (Mantel correlation; r = 0.478) only used in the non-migratory residential group provided support for the isolation by distance hypothesis.

Accordingly, the subsequent analyses of the 18 total localities including juveniles resulted in an enormous genetic divergence, suggesting that there were at least two widely differing populations depending on the allelic frequencies of *GPI-A* locus (Appendix A). However, they were provisionally categorized into three arbitrary groups, as shown on the map in Figure 1 (Japan, the East China Sea, and Taiwan inshore waters). The UPGMA trees based on allozyme markers on Figure 2a show four clades: A, B, C, and D. Clade A comprises the fish captured from Southern Taiwan: Tapong (TP), Kaoping (KP), and Peimen (PM). Clade B, located in the Northern Taiwan, includes Tadu (TD), Tansui (TS), Dashi (DS), and Hualien (HL). Clade C only comprises the sample from the Japanese coast (Nagasaki, NA). Finally, Clade D comprises samples from the coastal waters near Taiwan—Kaohsiung (KS), Chiding (CD), Anping (AP), and Wuchi (WC)—and off the coast of mainland China—Matsu (MS), Tachen (TC), and Shanghai (SH)—all of which were caught by gill nets on their way southbound from nearby coastal waters. These migratory populations partially mix with residential populations in midwinter during the migratory population’s annual southward movement. An NJ tree, constructed with Cavalli-Sforza and Edward’s chord distance models (Figure 2b), clearly shows that the Japanese NA sample clustered closer to the offshore migratory types than to the inshore residential type. Once the samples with the *GPI-A* locus were removed from the data, the position of NA became more tightly clustered with DS (Dashi, northern Taiwan) than to others (Figure 2c). This suggests that the *GPI-A* locus plays an important role in population differentiation.

PCoA was performed using the first two principal coordinates to investigate the population patterns using the genetic distances among samples. The variances in the first and second principal components were 77.46% and 13.24%, respectively, summing to 90.70% in total variation. PCoA indicated that the samples of *M. cephalus* could be split into three groups (Figure 3): fish from (1) PM, KP, TP, LBJ, TS, TD, DS, HL, TSJ, and FLJ; (2) NA; and (3) SH, TC, MS, WC, AP, CD, and KS. We then used the genetic STRUCTURE clustering algorithm previously adopted from allozyme data; the results indicated the presence of three distinct genetic clusters (K = 3) (Lnp(D) = −5730.13)) according to the ΔK metric developed by Evano et al. [35] (Figure 4a), which could test if the *GPI* locus and some other metabolic genetic loci might affect the population structuring. Figure 4b reveals three groups when all the loci remained, with four groups appearing once the *GPI-A* locus was removed. Similarly, the following LOSITAN test shown in Figure 5 could easily identify the outlier allozyme locus as *GPI-A*. Furthermore, BayeScan analysis indicated that *GPI-A* locus was under selection. For *M. cephalus*, the above 15 samples displayed a normal L-shaped allele frequency distribution in the mode-shift indicator, which indicated a stable population. None of the samples appeared have the significant heterozygosity excess that was observed by the Wilcoxon test under IAM, indicating that genetic bottlenecks were not detected in *M. cephalus* due to mutation-drift equilibrium. Therefore, the studied population of *M. cephalus* did not experience a recent genetic bottleneck.

### 3.2. Selection on GPI and PGDH

The overall offshore migratory samples (AP, CD, WC, MS, SH, TC, and KS) revealed had strikingly greater *F*_ST_ values on the *GPI* (0.489) and *PGDH* (0.202) loci (Appendix A). In addition, the frequency of *GPI-A*100* and *PGDH*100* decreased significantly between localities along with altitude (Figure 6, Appendix A); such a directional change in allelic frequencies was not found in other loci. As mentioned previously, the heterozygous genotype of *GPI-A*100/135* alone in all juveniles (27.89%) was 1.5 times higher than that of residential adults (18.96%) (Appendix A). To clarify the differences between juveniles and adults, all samples were pooled to determine if the proportion of *GPI-A*100/135* decreased with age. We found that the *GPI-A*100/100* increased in older fish, while *GPI-A*135/135* remained steady at all ages. The proportion of *GPI-A*100/135* decreased from larvae (27.89%) to 4-year-old adults (5.7%), and the *GPI-A*100/100* increased from larvae (20.20%) to 4-year-old adults (37.1%). This suggests that a negative effect of isozyme heterozygosity (*GPI-A*100/135*) may be maintained by selection (Figure 7).

## 4. Discussion

### 4.1. Genetic Diversity and Population Differentiation in Mugil cephalus 

Isozymes are commonly used genetic markers that provide additional information about the genetic structure of the species. The present study used a 13-polymorphic isozyme system to detect genetic variability in *M. cephalus*. The genetic diversity changes in time and space within species have been recognized as a fundamental part of biodiversity conservation, enabling populations to evolve in response to environmental changes [44]. Previous studies have found that populations with higher heterozygosity at locus-encoding enzymes tend to have significantly less fluctuating asymmetry and provide evidence that more heterozygous individuals within random mating populations are more developmentally stable [45]. The average heterozygosity (0.025) calculated from 21 isozyme loci for 15 local samples was far lower than that estimated for 10 samples from around the world [46] and Florida [47]. This is similar to the results from other animals—e.g., giant tiger prawn (0.027; range 0.018–0.046) [48]—but lower than those of the inshore sparid fish *Acanthopagrus schlegeli* (0.066; range 0.059–0.082) [27].

Levels of heterozygosity vary with ecological adaptability. Marine fish are generally found to have heterozygote deficiencies, which is probably caused by the occurrence of rare homozygous genotypes, inbreeding, selection, and the Wahlund effect [49]. The present investigation suggests that, for *M. cephalus*, overfishing is also an explanation for these deficiencies. Since 1986, catches of *M. cephalus* have sharply dropped due to intense fishing [50]. The previous study showed that *M. cephalus* experienced several demographic crashes due to decreases in surface water temperatures [18]. Although, *M. cephalus* did not exhibit a recent genetic bottleneck by isozyme markers. In the inshore large-scale mullet *Liza macrolepis*, the Ho values for the populations in the enclosed lagoon and estuaries (0.043–0.044) were much higher than those from open coasts (0.028–0.029) [51]. Populations with high gene flow also have a lower HE than those living in restricted areas. *Mugil cephalus* in the waters surrounding Taiwan was primarily subdivided into migratory and residential populations using *GPI-A* genotypes as a distinguishable criterion 18, with Ho values of 0.022 and 0.037, respectively. The wide difference is presumably due to the residential population being isolated, leading to low gene flow. In marine fish, loss of genetic diversity in wild populations has been reported in the northwest Pacific Ocean: the effective population size was reduced due to overfishing by commercial fisheries (e.g., Chinese pomfret, *Pampus chinensis* [52]; cutlassfish, *L. savala* [8]). The low heterozygosity values and significant number of polymorphisms found in grey mullets from the offshores of KS, CD, WC, AP, MS, SH, and TC (i.e., the migratory population) may be explained either by genetic drift, the founder effect, or strong directional selection due to geographic isolation (Appendix A). According to the pairwise *F*_ST_, higher population differentiation was detected between the migratory and residential populations (Table 3). We argue that this is probably caused by restricted gene flow between the migratory and residential populations. 

In general, population differentiation in *M. cephalus* may also be due to poor dispersal ability (Figure 3). This pattern is well supported by the results of the STRUCTURE analysis, in which these two groups or populations showed different patterns under K = 2 (Figure 4). Previous studies suggested that *M. cephalus* in the northwest Pacific Ocean is divided into two or three groups based on mtDNA and microsatellite (msat) loci [18,53]. The migratory and residential populations in our results resemble the NPW1 and NWP2 lineages, respectively, set by Shen et al. [18]. Our results show reduced genetic diversity in migratory populations. The Pleistocene climatic oscillations were likely less severe in southern latitudes, where the Kuroshio Current remained a stabilizing influence [18]. This aligns with previous studies suggesting that NWP2 has higher genetic diversity than NWP1, based on mtDNA COI gene analysis [18,53]. Many previous studies suggested that the major drop in sea level in the Taiwan Strait acted as a biogeographic barrier during major falls in sea level during the Pleistocene, which might have cut off migration on either side of the strait, forming two lineages in marine animals—e.g., the genera *Helice* [54], *B. sinensis* [6] and *L. savala* [8]. The results of the AMOVA test based on three geographical groups (Scenario II) indicate that there are significant differences among groups, suggesting a differentiation between migratory and residential populations. 

During the last glacier maximum (LGM) in the late Pleistocene, the Taiwan Strait acted as a geographical barrier and presented two lineages for marine fishes due to the descending sea level (e.g., *L. savala* [8]; *T. nanhaiensis* [55]). We suggested that migratory and residential populations probably diverged after the last glacial event. Growth rates differ slightly between these two populations. Temperature used to be considered one of the variables that affects growth, and growth may increase with rising temperatures in other species. However, a higher growth rate in migratory populations than in residential populations would appear to contradict the slower growth rate in species with a migratory population growing in habitats with lower temperatures. Since changes in allelic frequencies are not temperature dependent [23], selection is the more likely explanation for why these two populations are separate. The tendency for the genotypic frequencies of *GPI-A*100/100* to increase with age implies that the migratory population is well adapted to its environments. On the contrary, decreasing frequencies of the heterozygous allele suggest that allozyme variation may result from directional selection occurring in *100/135* in older fish, indicating strong selection within the residential population (Figure 6).

There are several selective mechanisms that can account for both widespread polymorphisms and geographic clines. One is divergent directional selection, with the intensity of selection varying in parallel with the relevant environmental variables along a geographic gradient [56]. According to LOSITAN and BayeScan, the outlier *GPI-A* locus (subject to positive directional selection) is a significant contributor to the genetic differentiation among populations; this suggests that natural selection (potentially resulting from genetic variation in environmental conditions) could be involved in driving genetic heterogeneity within *M. cephalus*. By itself, such a mechanism should lead to monomorphism at the geographic extremes. Our results reveal that the frequency of the *GPI-A*100* allele declined with decreasing latitude, and only the migratory populations (i.e., AP, CD, MS, WC, SH, TC, and KS) contained the *GPI-A*100* allele. Directional shifts in the allelic frequency of the *GPI-A* locus observed here suggest that *GPI-A*100* is favoured in higher-latitude habitats, while *GPI-A*135* is favoured in lower-latitude habitats. Thus, we argue that sea water temperature correlated with decreasing latitude might have promoted patterns of local adaptation and, therefore, genetic differentiation among *M. cephalus* populations (Figure 6). *PGDH* is the only allozyme locus in which this firm connection between latitudinal differences in the ambient temperature and allelic frequency has been found. The results of the AMOVA test based on the Taiwan, Japan, and mainland China groups when *GPI* loci were removed indicate that there were no significant differences at the group level. In general, when a single locus shows a pattern of variation that is highly discordant with other loci, it is likely that natural selection is acting on that locus [57]. Previous studies on fishes and other organisms have proposed that *PGI* (e.g., *GPI*) is a key enzyme for temperature adaptation [23,58]. Population genetic studies have revealed unusual patterns of variation at *PGI* in a wide range of organisms [59,60,61]. Temperature affects PGI allozyme functional properties such as the Michaelis–Menten binding constant, Km, and thermal stability in a diverse array of species [52,62]. In the grey mullet, the *GPI-A*100/100* homozygote exhibits a lower Km value and thermal stability compared to the *GPI-A*135/135* homozygote, suggesting that the *GPI-A*100* allele may function better than the *GPI-A*135* one at low temperatures. A similar case in a montane insect was reported by Dahlhoff and Rank [63]. It is proposed that the major role that *GPI* plays in glycolysis is especially critical for temperature adaptation [64].

Like with *GPI*, the *PGDH* data in this report demonstrate widespread polymorphism and a latitudinal cline. However, *PGDH* has fewer polymorphisms than *GPI* (Figure 6), but otherwise has a much less pronounced latitudinal cline. The *PGDH* frequency was observed from 0.94 to 0.99 in the migratory population (AP + CD + WC + KS), MS, TC, and SH to about 0.67 in TP. Therefore, while data support a hypothesis that the selective gradient is able to maintain a cline in the longitudinal direction for *PGDH*, it may be argued that clinal variation in the relative fitness of the homozygotes is more important than that of *GPI*. Isozyme loci that significantly deviate from neutral expectations of differentiation, or outlier loci, may be closely linked to the target of natural selection or even be directly under selection themselves [65]. The results of the present analyses clearly support the hypothesis that the latitudinal clines for *GPI* and *PGDH* are caused by negative correlations between the latitude and the frequencies of the *GPI-A*100* and *PGDH*100* alleles (Figure 6).

### 4.2. Genetic Variation among the Migratory Groups

The samples collected from the coastal waters of Nagasaki (NA) were similar to offshore samples collected from Taiwan, indicating a closer affiliation between migratory populations from either the coast of mainland China or Taiwan Island proper than to those from the inshore waters of Taiwan (Figure 2b). The same results showed that the Nagasaki samples were unique to those from other locations with the migratory element caught exclusively in the midwinter. In fact, the Nagasaki samples consisted of 17% *GPI-A 100/100*, 46% *GPI-A 130/130*, and 34% *GPI-A 100/130* hybrids and 2.5% *GPI-A 117/130* hybrids (Appendix A). These two hybrid forms had never been found in migratory populations leading toward the south. The above samples with a *GPI-A 100/100* locus among the annually migratory stocks were only slightly different from those with the same locus that resided in the inshore waters, per the *F*_ST_ (0.16042) and percent variation (3.18%) (Table 3VII). We found that some annual migrants settled in the inshore waters, because hybrids were found inside the near shore or estuaries. *GPI-A 100/100* and *GPI-A 135/135* alleles were used to delimit the above two populations, and Figure 2b shows that *GPI-A 100/100* and *GPI-A 135/135* had a surprisingly closer affiliation to the same inshore samples—e.g., *TP* with *100* allele vs. *TPR* with *135* alleles—than to the typical southward migratory schools carrying *GPI-A 100/100* alleles. A minority of the above migratory schools might have colonized the inshore waters along with the existing residential population. They did not return to the traditional nursery ground farther north. Nevertheless, the higher *F*_ST_ (0.57504) and 48.75% variation reported earlier (Table 3V) illustrated a striking separation between these two populations.

To confirm that the *GPI-A* locus dominates the changes in population structuring, a new, reorganized dataset was treated by removing all the samples with the entire locus. The new analysis resulted in a much lower *F*_ST_ (0.12855) and 1.34% variation (Table 3VI), which could only shallowly distinguish these two populations. The population structure of 15 samples with the lower *F*_ST_ (0.38472) and 18.70% variance was obtained once *GPI-A 100/100* and *GPI-A 135/135* were pooled (Table 3IV). The subsequent UPGMA tree newly constructed with Cavalli-Sforza and Edward’s chord distance models did not quite match with the above fishing localities, as indicated in Figure 2a, suggesting that intermingling might happen between migratory and residential populations. Further comparing the pairwise *F*_ST_ and AMOVA test results among 22 samples—including migratory *GPI-A 100/100* and the residential *GPI-A 135/135* genotypes—revealed significant differences: a higher variance of 48.75% and *F*_ST_ of 0.57504 compared to the variance of 18.707% and *F*_ST_ of 0.38472 obtained from the undivided 15 samples above (Table 3IV, V).

### 4.3. Genetic Variation among Non-Migratory Populations

The non-migratory residential group around the inshore waters of Taiwan is designated as the fish with *GPI 135/135* locus. A comparison between the semi-enclosed lagoon at Tapong and six other inshore sites (Kaoping, Peimen, Tadu, Tanshui, Dashi, and Hualien) yielded an *F*_ST_ of 0.103–0.239. TP is geographically closer to KP (*F*_ST_ = 0.103) than to the other sites (0.192–0.239). These results resemble those expressed in genetic distance. The population sub-structuring and dynamic models of grey mullets living in the inshore waters of Taiwan were exemplified by the semi-enclosed lagoon, Tapong Bay, in SW, Taiwan, where the fish reside year-round. The grey mullets in that lagoon consist of six *GPI-A* genotypes, predominately *135/135* (60.7%), followed by *100/135* (16.7%), *100/100* (16.1%), *117/135* (4.6%), *100/75* (1.1%), and *100/117* (0.74%). Their ages ranged from 0 to 5 years. Within the lagoon, the fish were only found in the age group 3–5 years during October and the following March (Figure 8A–C), indicating that the spawning ground was localized in the lagoon proper and probably extended into adjacent areas. The composition of the population in the lagoon was predominated by genotypes *GPI-A*135/135* (60.7%) and *GPI-A*100/135* (16.7%), summing to a total of 77.4% during the migration period and 82.83% during the non-migration period; this indicates that fish stocks moved slightly into the lagoon for feeding. On the other hand, the slight increase in the *GPI-A*100/100* genotype from 17.17% to 22.46% from moving inshore toward the lagoon represents the settlement of partially solitary migrating schools in the lagoon. Among the residential groups distributed in the seven localities above, the sample from KP is the most highly diversified with the remaining samples (*F*_ST_ 0.452–0.692), except TP (0.103).

### 4.4. Occurrence of Juvenile Grey Mullet

The occurrence of juveniles corresponded to the spawning seasons of the populations they belonged to, e.g., October–January for the residential population and December–March for the migrating population. The earliest catchable residential juveniles attained were 15 mm Ls in the Tanshui estuary and 90 mm Ls in the Linbien estuary near the outskirt of Tapong Bay, and they subsequently disappeared from the estuaries by May. The smallest migratory juveniles (20–25 mm Ls) with the *GPI-A*100/100* genotype caught in November grew to a peak size of 55–60 mm Ls in April in the Linbien estuary and then had disappeared by May. The large-sized residential population suggests that the residential population spawned earlier than the migratory population. Some juveniles with the migratory genotype (*GPI-A*100/100*) appeared in November and December and were partially produced by interbreeding with the heterozygous (*GPI-A*100/135*) residential population. A total of 1470 juveniles contained seven *GPI-A* genotypes—*100/100*, *100/135*, *135/135*, *100/117*, *135/117*, *100/75*, and *135/75*—with the following percentage compositions: 20.2%, 27.89%, 49.32%, 0.07%, 2.11%, 0.20%, and 0.20%, respectively (Appendix A). The genotypes with *135* alleles made up 79.32% of the composition and 64.42% of the allelic frequency. The latter was close to that of residential adults (63.26%). However, the heterozygous genotype of 100/135 alone in all juveniles (27.89%) was 1.5 times greater than the residential adults (18.96%), a result of selection in the course of growth (Appendix A).

### 4.5. Selection during Life Stages

Selection is key to the evolutionary process and has already been invoked to explain the differentiation in genetic variability (detected with allozymes) among marine populations from heterogeneous environments [66]. The proportion of the *GPI-A*100/135* heterozygous genotype in juveniles was higher than that in adults (27.89% vs. 18.96%). Therefore, data on all residential populations were pooled to test whether selection can act at a specific stage in time or is simply a result of a continuous process throughout the organism’s entire life span. The present analysis shows no selection that acts at a specific stage. Genotypic frequencies in juvenile samples closely match those observed for later age 0 juveniles. Furthermore, the genotypic frequencies change across life stages (Figure 7), suggesting that selection occurs throughout the life span.

In general, genetic polymorphisms being maintained in natural populations suggest that a population has adapted to environmental heterogeneity [67]. This process does not depend on local populations adapting to their environment. If selection favours a homozygous genotype for one type of environmental parameters and other homozygotes in another, then the heterozygous genotype may represent the highest arithmetic or harmonic mean fitness [68]. Thus, the heterozygous state may act as a buffer against environment variation, and the presence of environmental heterogeneity results in the realization of a higher heterozygote fitness potential [69]. However, our results differ from this description in that the heterozygous genotype acts against a strong selection force.

### 4.6. Linkage Disequilibrium between GPI-A and PGDH

Linkage disequilibrium and non-random associations between alleles or groups of nucleotides may indicate epistatic relationships. The pattern of pairwise linkage disequilibrium is consistent between populations, indicating that strong evidence of selection drives the extent of linkage disequilibrium (LD) [70]. Linkage disequilibrium between individual locus is very scarce in fish, except when genes extremely closely linked or chromosomal inversions are often associated. When significant associations have been observed, it is often not at all clear whether they are due to non-random sampling of haplotype, random genetic drift, or natural selection. Significant disequilibrium can indeed arise without epistasis from random genetic drift within a given population in subdivided populations and by gene migration or the founder effect [71]. In Drosophila, numerous examples of significant linkage disequilibrium have been discovered between specific allozyme and chromosomal inversions, which have been explained as reflecting selection for favoured multilocus allelic combinations. In general, linkage disequilibrium is mostly associated with closely linked genes but may involve distantly linked genes when special cytological mechanisms allow it to exist [72]. We do not know the positions of the *GPI-A* and *PGDH* genes in the chromosome. The significant linkage disequilibrium of *GPI-A* and *PGDH* genes identified in the present study is presumably a result of special cytological mechanisms. However, it remains unclear what combination of environmental variables could result in a decrease in the frequency of the *GPI-A*100/135* genotype. The proportion of *PGDH*100/100* slightly declined with age, but *PGDH*125/125* otherwise increased with age (Figure 7). The proportion of *PGDH*125/125* increased from larvae (9.2%) to 4-year-old adults (25.7%) The genotypic frequency distribution patterns were similar between *GPI-A*100/135* and *PGDH*100/100* (Figure 7). The distinct selection regime of environmental stresses cannot be fully explained. Further examination of biochemical and physiological properties of the *GPI* and *PGDH* genotypes in grey mullet under a variety of environmental conditions may help resolve this problem.

## 5. Conclusions

Our results indicate that all populations are clustered into three distinct genetic groups in the PCoA analysis. In this study, migratory and resident populations were classified as the NWP1 and NWP2 lineages, respectively, based on isozyme data. The differences between migratory and residential populations suggest that the major drop in sea level in the Taiwan Strait acted as a biogeographic barrier during major falls in sea level in the Pleistocene, when the fish’s migration route was cut off on either side of the strait to form two lineages of marine fishes. *Mugil cephalus* was detected in the migratory and residential populations with Ho values of 0.022 and 0.037, respectively. The wide difference is presumably due to the residential population being isolated, leading to low gene flow. AMOVA and the high overall mean *F*_ST_ of 0.252 indicated enormous genetic differentiation among populations. The significant contribution of the outlier *GPI-A* locus (subject to positive directional selection) to the genetic differentiation among populations based on LOSITAN and BayeScan suggests that natural selection (potentially resulting from genetic variation in environmental conditions) could be involved in driving genetic heterogeneity within *M. cephalus*. Heterozygote deficiencies and the loss of the genetic diversity found in *M. cephalus* are probably the results of overfishing by commercial fisheries. During the life history of *M. cephalus*, the *GPI-A*100/135* heterozygote frequency significantly decreases with age. Based on these data, we suggest that each *GPI-A* genotype represents trait combinations of higher fitness in some portions of the environment. While this study offers valuable insights into the genetic differentiation of *Mugil cephalus* populations, several key gaps remain that need to be addressed to strengthen the findings and guide future research. These include limited geographic sampling and a sufficient focus on environmental variables.

## Figures and Tables

**Figure 1 genes-15-01280-f001:**
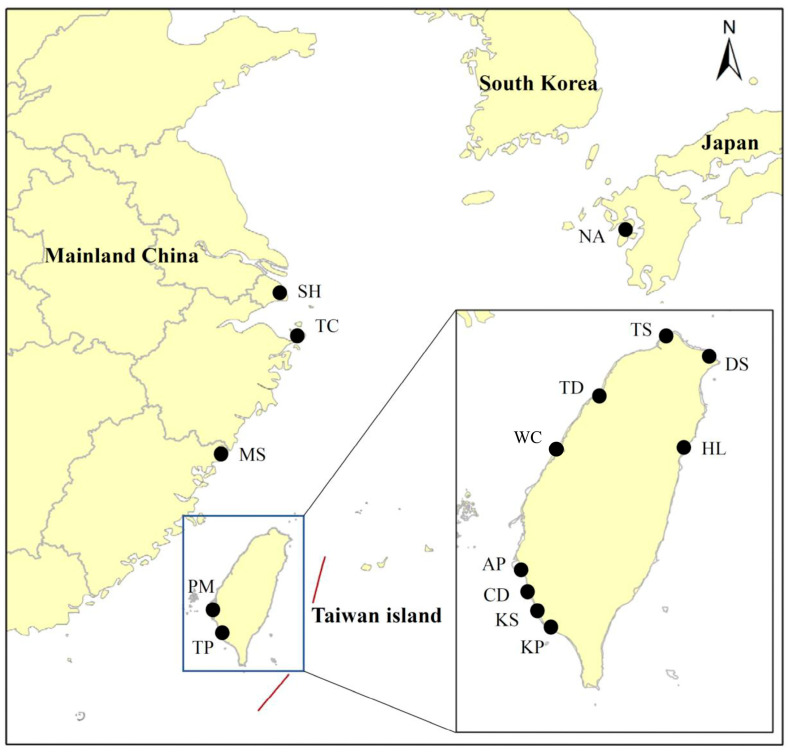
Map showing the 18 sampling localities of *Mugil cephalus*. Collection sites (circles) correspond to locations given in the text and Table 1. The map showing the 18 sampling localities of *Mugil cephalus* © 2022 by Hung-Du Lin is licenced under Attribution 4.0 International. The map was created using the Microsoft Paint app in Windows 10.

**Figure 2 genes-15-01280-f002:**
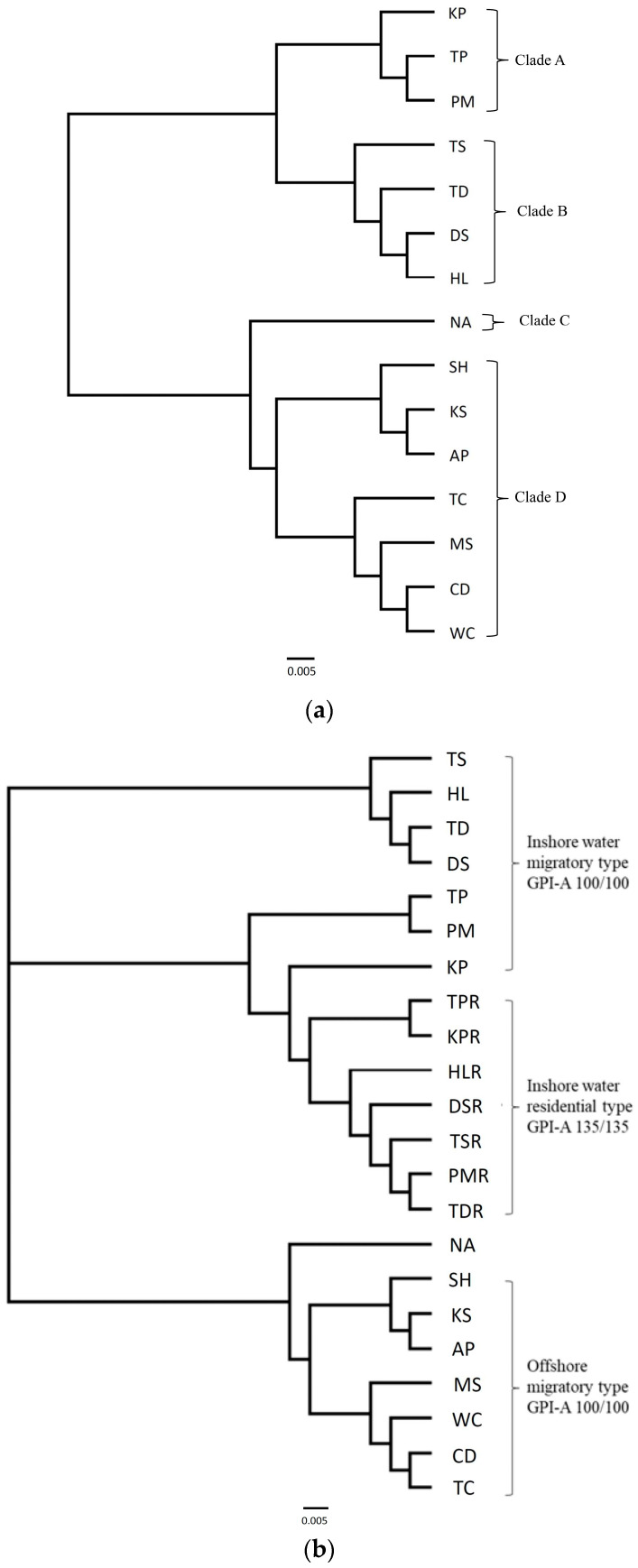
(**a**) UPGMA dendrogram of pooled *Mugil cephalus* collected among 15 localities based on Cavalli-Sforza and Edward’s chord distance. AP, Anping; CD, Chiding; WC, Wuchi, TC, Tachen; DS, Dashi; HL, Hualien; KP, Kaoping estuary; KS, Kaohsiung; MS, Matsu; SH, Shanghai; NA, Nagasaki (Japan); PM, Peimen; TD, Tadu; TP, Tapong; TS, Tamshui. (**b**) The NJ tree constructed with Cavalli-Storza and Edward’s chord distance models. The *GPI-A 100/100* and *GPI-A 135/135* genotypes were independently counted on the inshore samples yielded. The letter R behind the localities indicates inshore samples. (**c**) A new NJ tree constructed using Cavalli-Sforza and Edward’s chord distance models when the entire *GPI-A* locus is removed.

**Figure 3 genes-15-01280-f003:**
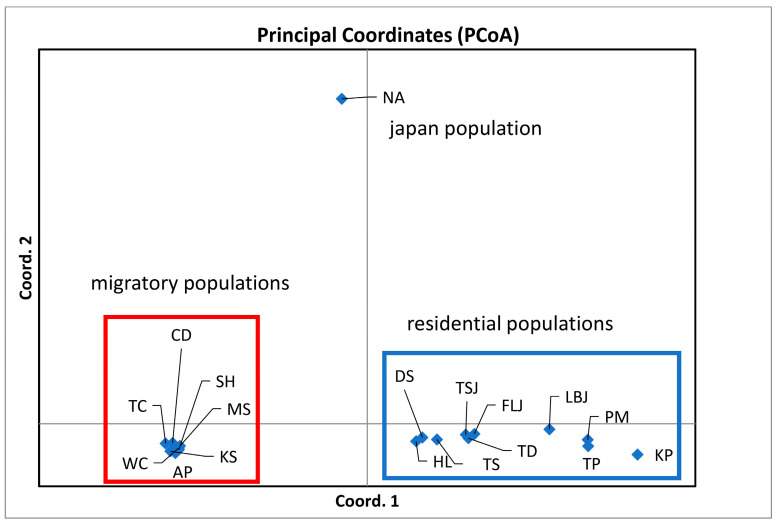
Principal component analysis (PCoA) of the *Mugil cephalus* populations in Japan, mainland China, and Taiwan based on the allozyme dataset. For each axis, eigenvalues, variance (% variance), and cumulative variance (% cum variance) are provided. The letter J behind the localities represents juveniles. NA: Nagasaki population in Japan.

**Figure 4 genes-15-01280-f004:**
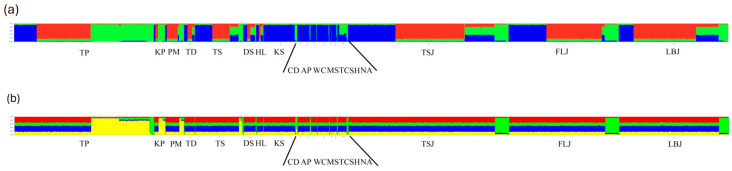
(**a**) Clustering of individuals using Structure at K = 3 (with all loci included). (**b**) Clustering of individuals using Structure at K = 4 (with the *GPI-A* locus removed). Individuals are represented by vertical bars, with each colour representing one cluster and the length of the coloured segment indicating the individual’s estimated degree of kinship to that cluster (Y-axis). The different colours correspond to the population designations given at the bottom of the figure (X-axis). Populations are separated by black bars, and abbreviation are defined in Table 1.

**Figure 5 genes-15-01280-f005:**
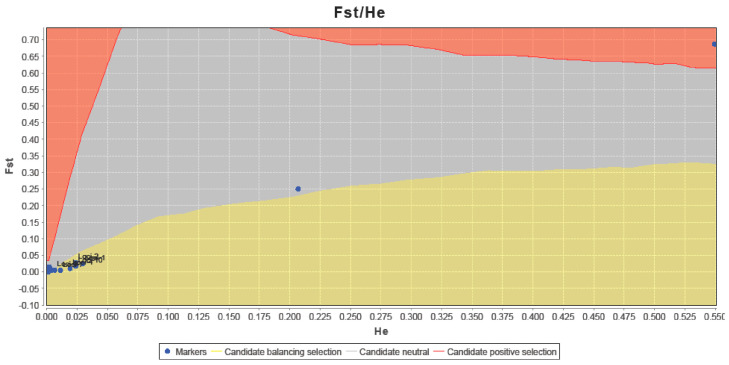
Test for neutrality using LOSITAN based on 13 allozyme loci. Observed value of heterozygosity vs. *F*_ST_ at each locus (black dots) and 95% confidence envelope expected under neutrality.

**Figure 6 genes-15-01280-f006:**
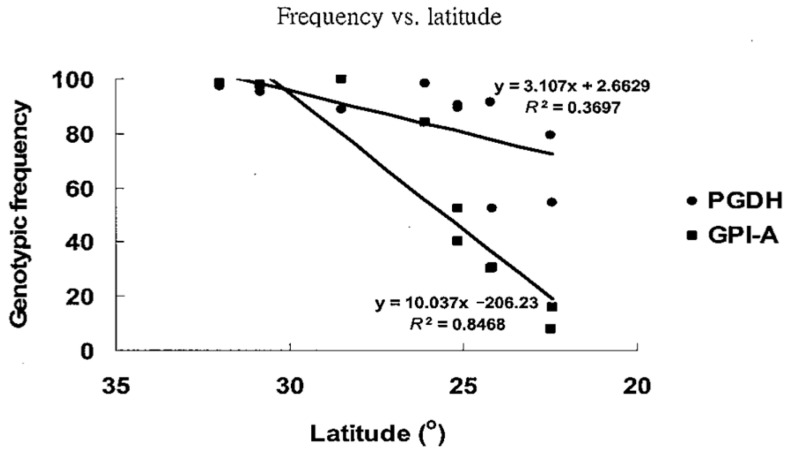
Frequencies of *GPI-A 100/100* and *PGDH 100/100* genotypes at different latitudes.

**Figure 7 genes-15-01280-f007:**
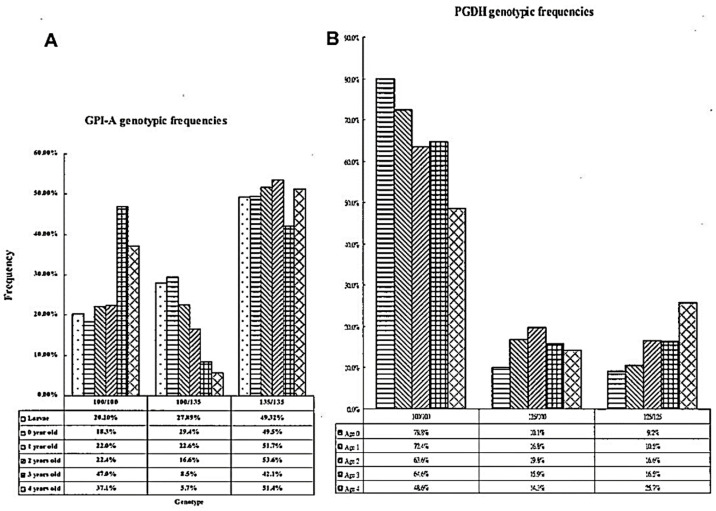
Genotype frequency distribution of *GPI-A* (**A**) and *PGDH* (**B**) over the grey mullet life history.

**Figure 8 genes-15-01280-f008:**
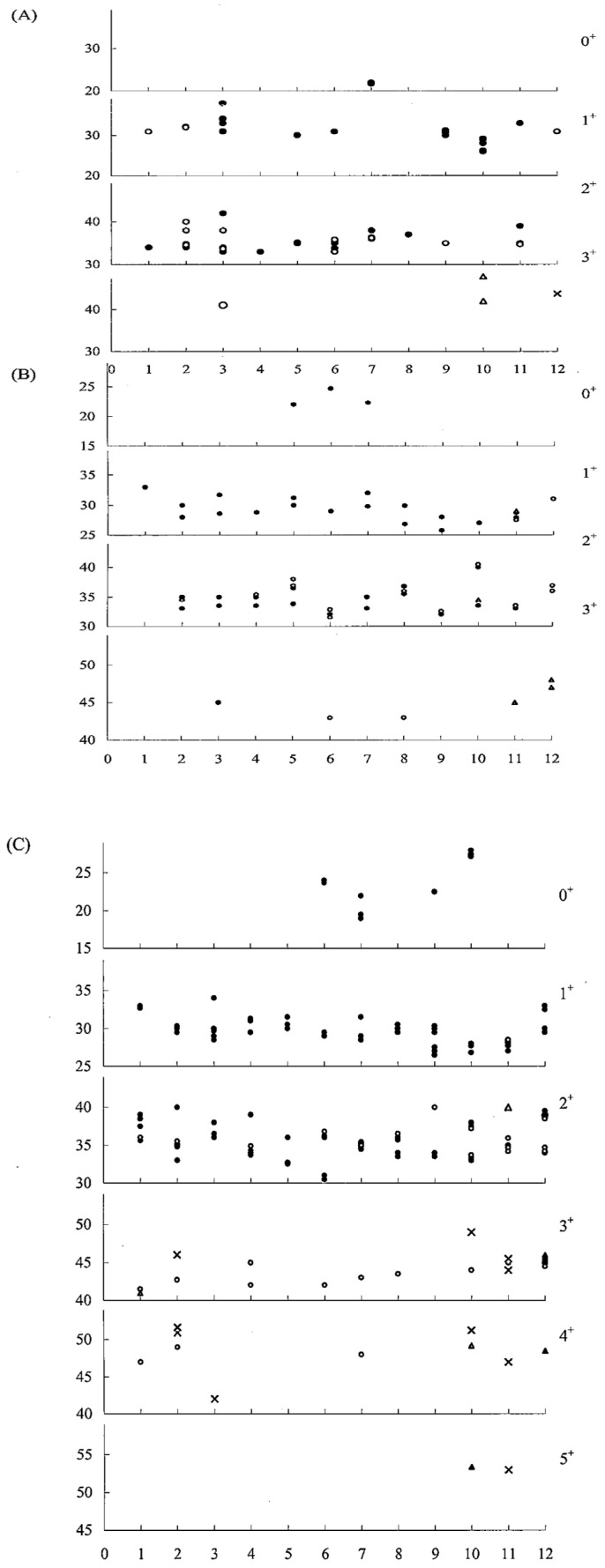
Gonad maturity stages in relation to body sizes and ages of Tapong grey mullets classified into *GPI-A 100/100* (**A**), *GPI-A 100/135* (**B**), and *GP1-A 135/135* (**C**) groups. Gonad stages: ●, I; ○, II; △, III; ▲, IV; ×, V. Note: The x-axis represents a period (monthly), and the y-axis shows body size (cm). 0^+^, 1^+^, 2^+^, 3^+^, 4^+^, 5^+^ represents the age.

**Table 1 genes-15-01280-t001:** Genetic diversity measures and the exact test for Hardy–Weinberg equilibrium in grey mullet *Mugil cephalus* samples, including some juveniles.

Population	Abbreviation	N ^a^	A ^b^	A_R_ ^c^	Ho ^d^	He ^e^	Fis ^f^
Adult							
Nagasaki	NA	41	2.500	1.259	0.164	0.198	0.174
Shanghai	SH	22	2.286	1.382	0.058	0.093	0.383
Tachen	TC	9	2.000	1.231	0.111	0.111	0.000
Matsu	MS	48	2.167	1.125	0.031	0.031	−0.007
Dashi	DS	50	2.000	1.201	0.132	0.143	0.083
Tanshui	TS	187	2.250	1.214	0.055	0.092	0.402
Wuchi	WC	25	2.000	1.119	0.066	0.065	−0.017
Tadu	TD	42	2.200	1.186	0.080	0.132	0.390
Peimen	PM	73	2.167	1.247	0.100	0.131	0.235
Anping	AP	31	2.000	1.141	0.053	0.092	0.423
Chiding	CD	20	2.000	1.035	0.050	0.050	0.000
Kaoshiung	KS	131	2.500	1.123	0.017	0.056	0.698
Kaoping	KP	44	3.000	1.207	0.215	0.442	0.515
Tapong	TP	540	2.333	1.283	0.049	0.083	0.407
Hualien	HL	28	2.000	1.155	0.142	0.215	0.340
Juvenile							
Fulung	FLJ	426	2.800	1.159	0.059	0.127	0.532
Tanshui	TSJ	621	3.400	1.153	0.076	0.121	0.367
Linbien	LBJ	423	3.000	1.160	0.099	0.129	0.231
Mean			2.367	1.188	0.086	0.128	

^a^ Number of specimens; ^b^ mean number of alleles; ^c^ mean allelic richness with standard deviation; ^d^ mean observed heterozygosity with standard deviation; ^e^ mean expected heterozygosity with standard deviation; ^f^ inbreeding coefficient.

**Table 2 genes-15-01280-t002:** Summary of *F*-statistics (*F*_IS_, *F*_IT_, and *F*_ST_) at 13 polymorphic loci for 18 populations of *Mugil cephalus*. The estimation was carried out according to Weir and Cockerham (1984).

Locus	N_a_ ^a^	A_R_ ^b^	H_O_ ^c^	H_E_ ^d^	F_IS_	F_IT_	F_ST_
mAAT	3	1.153	0.007	0.028	0.643	0.652	0.025
CK-A	2	1.102	0.024	0.023	−0.031	−0.002	0.029
GPI-A	5	2.415	0.185	0.296	0.454	0.570	0.214
GPI-B	4	1.481	0.031	0.031	−0.029	−0.018	0.011
IDH-A	2	1.007	0.001	0.001	−0.002	0.000	0.003
IDH-B	2	1.019	0.008	0.008	−0.016	0.002	0.017
LDH-A	4	1.023	0.013	0.013	−0.021	0.003	0.023
LDH-B	2	1.032	0.004	0.004	−0.005	−0.000	0.004
MDH-A	2	1.003	0.000	0.000	0.002	−0.000	−0.003
MPI	4	1.09	0.026	0.025	−0.016	−0.001	0.015
PGM-A	4	1.055	0.003	0.003	−0.002	−0.002	−0.000
PGM-B	2	1.026	0.001	0.002	0.246	0.250	0.004
PGDH	3	1.84	0.070	0.123	0.440	0.582	0.253
Jackknifing over loci							
Total					0.452 ± 0.124	0.578 ± 0.130	0.218 ± 0.044
Bootstrapping over loci (95% confidence interval)							
					−0.008 ± 0.447	0.007 ± 0.564	0.012 ± 0.223
Bootstrapping over loci (99% confidence interval)							
					−0.024 ± 0.453	−0.012 ± 0.569	0.011 ± 0.233

^a^: Number of alleles; ^b^: Allelic richness; ^c^: Observed heterozygosity; ^d^: Expected heterozygosity.

**Table 3 genes-15-01280-t003:** Analysis of molecular variance (AMOVA) for 15 *Mugil cephalus* populations based on allozyme loci.

Scheme Category Description	% Var.	Statistic	*p*
One geographical group			
Among populations within groups	19.30	*F*_ST_ = 0.193	0.000
Within populations	80.70		
Scenario I: Three geographical groups (Taiwan, Japan, and mainland China)			
Among groups	27.20	*F*_SC_ = 0.163	0.000
Among populations within groups	11.90	*F*_ST_ = 0.390	0.000
Within populations	60.91	*F*_CT_ = 0.271	0.009
Scenario II: Three geographical groups (NA); (WC, CD, KS, AP, SH, TC, and MS); (TS, DS, TD, HL, KP, PM, TP, TSJ, FLJ, and LBJ)			
Among groups	34.71	*F*_SC_ = 0.086	0.000
Among populations within groups	5.66	*F*_ST_ = 0.403	0.000
Within populations	59.64	*F*_CT_ = 0.347	0.000
Scenario III: Four ecological groups (NA); (WC, CD, KS, AP, SH, TC, and MS); (TS, DS, TD, HL, KP, PM, and TP); (TSJ, FLJ, and LBJ)			
Among groups	19.85	*F*_SC_ = 0.056	0.000
Among populations within groups	4.53	*F*_ST_ = 0.243	0.000
Within populations	75.62	*F*_CT_ = 0.198	0.000
Scenario IV: Three geographical groups (Taiwan vs. other populations)			
Among groups	18.70	*F*_SC_ = 0.243	0.000
Among populations within groups	19.77	*F*_ST_ = 0.384	0.000
Within populations	61.53	*F*_CT_ = 0.186	0.000
Scenario V: Residential and migratory, among 22 samples when GPI 100/100 and GPI 135/135 are treated separately			
Among groups	48.75	*F*_SC_ = 0.170	0.000
Among populations in group	8.75	*F*_ST_ = 0.575	0.000
Within population	42.50	*F*_CT_ = 0.487	0.000
Scenario VI: Taiwan, Japan, and mainland China groups when GPI loci are removed			
Among groups	1.34	*F*_SC_ = 0.116	0.000
Among populations in group	11.52	*F*_ST_ = 0.128	0.000
Within population	87.15	*F*_CT_ = 0.013	0.000
Scenario VII: Taiwan and Japan migratory groups			
Among groups	3.18	*F*_SC_ = 0.132	0.000
Among populations in group	12.86	*F*_ST_ = 0.160	0.000
Within population	83.96	*F*_CT_ = 0.031	0.000

## Data Availability

The dataset was deposited in Appendix A. Voucher specimens are housed at the Institute of Cellular and Organismic Biology, Academia Sinica, Taiwan.

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
