# Peer review of "Variation in the Local Grey Mullet Populations (Mugil cephalus) on the Western Pacific Fringe"

_genes, 2024, doi:10.3390/genes15101280_

Round 1

Reviewer 1 Report

Comments and Suggestions for Authors

Dear authors,

Congratulations for your work results.

Please find below some suggestions.

1. Abstract. The grey mullet (Mugil cephalus) is a globally distributed 17 coastal fish. please be more acurate in by changing this statement. in introduction is ok.

2. Introduction. Complex geological processes and climate history in different parts of the world can 43 help shape the present phylogeographic patterns of organisms in the marine environment. in my opinion there are other causes too which participate to this shapings. please add them.

3. All the not original texts in the paper should have citations and references at the end of the paper. only one example is the upper phrase, but not only in the whole text.

4. Several recent studies have shown that significant genetic barriers between two evolu- 51 tionary lineages of marine species due to the behavioral or oceanographic constraints, 52 such as in Chinese black sleeper (Bostrychus sinensis) [1], and shimofuri goby (Tridentiger 53 bifasciatus) [2], and low genetic structure of marine species due to higher migration rates, 54 such as in cutlassfish (Lepturacanthus savala) [3], prawn (Macrobrachium japonicum) [4], and 55 hairtail (Trichiurus japonicus) [5]. hard to understend what you mean ..... may be some word/words missing ...

5. May be a synthetic phrase about how the studied species biological and ecological characteristics influence the genetic potential trends can help the reader to understend better the importance of the study, after the 59-82 rows paragraph.

6. Highlight more the need and importance of your study.

materials and methods 7. highlight the limitation of the approach and methods used.

Discussion 8. conservation, enabling populations to evolve in response with environmental changes 38 brackets missing here around 38

9. Our results revealed that genetic diversity 391 was lower in migratory populations. present migratory populations? it is somehow counter intuitive. you can explain this?

10. Previous studies on insects and other organisms have proposed that PGI (e.g., GPI) is a 437 key enzyme for temperature adaptation, etc ..... better to use fish cases for comparison goals

Conclusions 11 stay close to your valuable work conclusions not go arround general info. ex: the grey mullet (Mugil cephalus) is a commercially important marine species

12 A short critical gap analysis of your work results can identify future directions in completing this study in the futire

reference 13. the references should be richer in my opinion.

All the best

Reviewer

Author Response

  1. Abstract. The grey mullet (Mugil cephalus) is a globally distributed coastal fish. please be more acurate in by changing this statement. in introduction is ok.

→As requested, we revised this sentence. “The grey mullet (Mugil cephalus) is one of the few marine coastal fish species with a global circumtropical distribution.”

  1. Introduction. Complex geological processes and climate history in different parts of the world can help shape the present phylogeographic patterns of organisms in the marine environment. in my opinion there are other causes too which participate to this shapings. please add them.

→As requested, we revised this sentence. “Complex geological processes, climatic history, and diverse coastal habitats across different regions of the world create opportunities that shape the current phylogeographic patterns of marine organisms.“

  1. All the not original texts in the paper should have citations and references at the end of the paper. only one example is the upper phrase, but not only in the whole text.

→As requested, we added the citations in the manuscript. For example, Complex geological processes, climatic history, and diverse coastal habitats across different regions of the world create opportunities that shape the current phylogeographic patterns of marine organisms [1, 2]. Ocean currents, ancestral habitat discontinuity, and climatic constraints in the Northwest Pacific may play an important role in shaping the contemporary genetic and population structures of marine organism [3]. During the Pleistocene glacial periods, the emergence of the Taiwan Strait due to the decline of sea levels might have acted as a barrier to the movement of marine organisms between the two sides of this strait [4, 5].

  1. Bowen, B. W., Gaither, M. R., DiBattista, J. D., Iacchei, M., Andrews, K. R., Grant, W. S., ... & Briggs, J. C. (2016). Comparative phylogeography of the ocean planet. Proceedings of the National Academy of Sciences, 113(29), 7962-7969.
  2. Conover, D. O., Clarke, L. M., Munch, S. B., & Wagner, G. N. (2006). Spatial and temporal scales of adaptive divergence in marine fishes and the implications for conservation. Journal of fish biology69, 21-47.
  3. Chiu, Y.W.; Bor, H.; Wu, J.X.; Shieh, B.S.; Lin, H.D. Population Structure and Phylogeography of Marine Gastropods Monodonta labio and M. confusa (Trochidae) along the Northwestern Pacific Coast. Diversity, 2023, 15, 1021. https://doi.org/10.3390/d15091021
  4. Liu, J.X.; Gao, T.X.; Wu, S.F.; Zhang, Y.P. Pleistocene isolation in the Northwestern Pacific marginal seas and limited dispersal in a marine fish, Chelon haematocheilus (Temminck & Schlegel, 1845). Mol. Ecol. 2007, 16, 275-288. https://doi.org/10.1111/j.1365-294X.2006.03140.x
  5. Chen, X.; Wang, J.J.; Ai, W.M.; Chen, H.; Lin, H.D. Phylogeography and genetic population structure of the spadenose shark (Scoliodon macrorhynchos) from the Chinese coast. Mitochondrial Dna Part A, 2018, 29(7), 1100-1107. https://doi.org/10.1080/24701394.2017.1413363

  1. Several recent studies have shown that significant genetic barriers between two evolutionary lineages of marine species due to the behavioral or oceanographic constraints, such as in Chinese black sleeper (Bostrychus sinensis) [1], and shimofuri goby (Tridentiger bifasciatus) [2], and low genetic structure of marine species due to higher migration rates, such as in cutlassfish (Lepturacanthus savala) [3], prawn (Macrobrachium japonicum) [4], and hairtail (Trichiurus japonicus) [5]. hard to understend what you mean ..... may be some word/words missing ...

→As requested, we revised these sentences. “During the Pleistocene glacial periods, the emergence of the Taiwan Strait due to the decline of sea levels might have acted as a barrier to the movement of marine organisms between the two sides of this strait [4, 5]. These historical geological events, combined with complex coastal habitats, have been linked to significant genetic barriers between two evolutionary lineages of marine species due to behavioral or oceanographic constraints, such as in Chinese black sleeper (Bostrychus sinensis) [6] and shimofuri goby (Tridentiger bifasciatus) [7]. Conversely, some species exhibit low genetic structure, likely due to higher migration rates, as seen in cutlassfish (Lepturacanthus savala) [8], prawn (Macrobrachium japonicum) [9], and hairtail (Trichiurus japonicus) [10].”

  1. May be a synthetic phrase about how the studied species biological and ecological characteristics influence the genetic potential trends can help the reader to understend better the importance of the study, after the 59-82 rows paragraph.

→As requested, we revised the sentences, splitting them into two paragraphs to enhance readability. The revision condenses certain details, maintains a smooth flow, and ensures clarity while preserving the original meaning. “The grey mullet (Mugil cephalus) is a globally distributed marine fish, inhabiting coastal waters and estuaries in tropical and subtropical seas between latitudes 42°N and 42°S [11, 12, 13]. Spawning occurs offshore near the surface, with buoyant eggs hatching approximately 48 hours after fertilization. Larvae are dispersed across the continental shelf by ocean currents, spending the first 2–3 months in a planktonic stage. During this phase, they grow to a standard length (Ls) of 16–20 mm and form dense schools that migrate towards inshore waters and estuaries [14]. Young recruits first appear in the surf zone before moving into the shallow areas of sounds, bays, and estuaries. Here, juveniles (40–69 mm Ls) spend their first year in waters with salinities ranging from 0 to 35‰ [15]. Adult grey mullets primarily feed on detritus and reach sexual maturity in their third year, at which point they form large migratory schools [16, 17].

In Taiwan’s coastal waters and estuaries, two types of grey mullet are observed: migratory and resident [18]. The migratory type originates from the northern East China Sea, traveling along mainland China’s coast to the Taiwan Strait during the spawning season [14]. Migratory adults and juveniles then return to the mainland coast, aided by the Kuroshio Current, while juveniles remain in Taiwanese estuaries until late April. In contrast, the resident type inhabits Taiwanese estuaries year-round, with minimal migration. The grey mullet's life history involves both passive and active dispersal mechanisms, primarily along the shoreline, which likely shape population subdivisions. Understanding the population dynamics and genetic structure of this species is crucial, given its significance to coastal fisheries and aquaculture in many countries worldwide [13].”

  1. Highlight more the need and importance of your study.

→As requested, we have revised the sentences and rewritten the highlight of our research. “Huang et al. [23] previously employed the GPI-A locus as a genetic marker to distinguish between at least two grey mullet stocks—migratory and residential—within the western Pacific fringe, indicating that natural selection might influence allelic variation. In this study, we applied allozyme electrophoresis to investigate the potential genetic structure of grey mullet populations around Taiwan. Our objectives were threefold: first, to assess genetic divergence and gene flow among western Pacific populations of Mugil cephalus; second, to evaluate whether local populations are subject to selective pressures; and third, to determine the timing of selection during the grey mullet's life cycle, specifically whether genotypic mortality occurs progressively or results from intense selection at a particular developmental stage.”

materials and methods 7. highlight the limitation of the approach and methods used.

→As requested, we have revised and added the sentence. “Allozyme analysis is a technique used in population genetics and evolutionary biology, but it has limitations, such as limited detection of genetic variability and susceptibility to environmental influences [29].”

  1. Bossart, J.L.; Prowell, D.P. Genetic estimates of population structure and gene flow: limitations, lessons and new directions. Trends Ecol. Evol. 1998, 13, 202-206. https://doi.org/10.1016/S0169-5347(97)01284-6

Discussion 8. conservation, enabling populations to evolve in response with environmental changes 38 brackets missing here around 38

→As requested, we have revised the sentence and added the necessary brackets.

  1. Our results revealed that genetic diversity was lower in migratory populations. present migratory populations? it is somehow counter intuitive. you can explain this?

→As requested, we have revised and added the sentence. “Our results showed reduced genetic diversity in migratory populations. The Pleistocene climatic oscillations were likely less severe in southern latitudes, where the Kuroshio Current remained a stabilizing influence [18]. This aligns with previous studies suggesting that NWP2 has higher genetic diversity than NWP1, based on mtDNA COI gene analysis [18, 53].”

  1. Previous studies on insects and other organisms have proposed that PGI (e.g., GPI) is a key enzyme for temperature adaptation, etc ..... better to use fish cases for comparison goals

→As requested, we have revised the sentence. “Previous studies on fishes and other organisms have proposed that PGI (e.g., GPI) is a key enzyme for temperature adaptation [58].”

  1. Huang, C., Weng, C., & Lee, S. (2001). Distinguishing two types of gray mullet, Mugil cephalus L. (Mugiliformes: Mugilidae), by using glucose-6-phosphate isomerase (GPI) allozymes with special reference to enzyme activities. Journal of Comparative Physiology B171, 387-394.
  2. Riddoch, B.J. The adaptive significance of electrophoretic mobility in phosphoglucose isomerase (PGI). Biol. J. Linn. Soc., 1993, 50, 1-17. https://doi.org/10.1111/j.1095-8312.1993.tb00915.x

Conclusions 11 stay close to your valuable work conclusions not go arround general info. ex: the grey mullet (Mugil cephalus) is a commercially important marine species

→As requested, we have deleted and revised the sentences. The original sentence: “The grey mullet (Mugil cephalus) is a commercially important marine species that is divided into two or three groups based on mtDNA and microsatellite (msat) loci in the northwest Pacific Ocean. PCoA analysis showed that all populations were grouped into three genetic clusters. In our study, migratory and residential populations were regarded as the NPW1 and NWP2 lineages, respectively, based on isozyme data.” was revised to: “Our results indicated that all populations clustered into three distinct genetic groups in the PCoA analysis. In this study, migratory and resident populations were classified as the NWP1 and NWP2 lineages, respectively, based on isozyme data."

12 A short critical gap analysis of your work results can identify future directions in completing this study in the futire

→As requested, we have added and revised the sentences. “While the study offers valuable insights into the genetic differentiation of Mugil cephalus populations, several key gaps remain that need to be addressed to strengthen the findings and guide future research. These include limited geographic sampling and an insufficient focus on environmental variables.”

reference 13. the references should be richer in my opinion.

→As requested, we have added more comprehensive and diverse references.

Reviewer 2 Report

Comments and Suggestions for Authors

I am very skeptical regarding this study. The study characterized a very large number of grey mullets from a wide area, resulting in interesting results at a population basis. Nevertheless there is a very big limitation of the study, i.e. that the authors only confirmed results from previous studies based on mtDNA and microsatellite analyses, and in fact, using a relatively old technique. Allozymes have been used in the past but have the disadvantage of examining only loci located on coding genomic sequence. This is the reason why they are not used very much at the time being. 

Minor comments

Figure 1a should have a different name than Figure 1b, c and d

Line 214. This is confusing as written. Please replace with “isozyme markers”

Author Response

Reviewer   2

Comments and Suggestions for Authors

I am very skeptical regarding this study. The study characterized a very large number of grey mullets from a wide area, resulting in interesting results at a population basis. Nevertheless there is a very big limitation of the study, i.e. that the authors only confirmed results from previous studies based on mtDNA and microsatellite analyses, and in fact, using a relatively old technique. Allozymes have been used in the past but have the disadvantage of examining only loci located on coding genomic sequence. This is the reason why they are not used very much at the time being. 

→We understand that the allozyme technique is relatively old and has its limitations. We have included a discussion of this in the article to help readers appreciate the contributions of this research.

As requested, we have revised and added the sentence. “Allozyme analysis is a technique used in population genetics and evolutionary biology, but it has limitations, such as limited detection of genetic variability and susceptibility to environmental influences [29].”, “While the study offers valuable insights into the genetic differentiation of Mugil cephalus populations, several key gaps remain that need to be addressed to strengthen the findings and guide future research. These include limited geographic sampling and an insufficient focus on environmental variables.”

Minor comments

Figure 1a should have a different name than Figure 1b, c and d

→As requested, we have labeled Figure 1a as Figure 1, and modified Figure 1b, c, and d to Figure 2a, b, and c.

Line 214. This is confusing as written. Please replace with “isozyme markers”
